# Use of Patient-Reported Data within the Acute Healthcare Context: A Scoping Review

**DOI:** 10.3390/ijerph191811160

**Published:** 2022-09-06

**Authors:** Kathryn Kynoch, Mary Ameen, Mary-Anne Ramis, Hanan Khalil

**Affiliations:** 1Mater Health and Queensland Centre for Evidence Based Nursing and Midwifery, A JBI Centre of Excellence, Brisbane 4006, Australia; 2Faculty of Medicine, Nursing and Health Sciences, Monash Rural Health Churchill, Monash University, Churchill 3842, Australia; 3School of Psychology and Public Health, La Trobe University, Melbourne 3086, Australia

**Keywords:** PREMs, PROMs, patient satisfaction, patient-reported data, patient-reported experience, scoping review

## Abstract

Patient-reported outcome measures (PROMs), patient-reported experience measures (PREMs) and patient satisfaction surveys provide important information on how care can be improved. However, data collection does not always translate to changes in practice or service delivery. This scoping review aimed to collect, map and report on the use of collected patient-reported data used within acute healthcare contexts for improvement to care or processes. Using JBI methods, an extensive search was undertaken of multiple health databases and trial registries for published and unpublished studies. The concepts of interest included the types and characteristics of published patient experience and PROMs research, with a specific focus on the ways in which data have been applied to clinical practice. Barriers and facilitators to the use of collected data were also explored. From 4057 records, 86 papers were included. Most research was undertaken in North America, Canada or the UK. The Hospital Consumer Assessment of Healthcare Providers and Systems tool (HCAHPS) was used most frequently for measuring patient satisfaction. Where reported, data were applied to improve patient-centred care and utilization of health resources. Gaps in the use of patient data within hospital services are noticeable. Engaging management and improving staff capability are needed to overcome barriers to implementation.

## 1. Introduction

Delivering individualized patient-centred care is a valuable characteristic of high-quality health care services. To address this goal, many organisations collect patient-reported experience measures (PREMs), patient-reported outcome measures (PROMs) and patient satisfaction questionnaires. These measures can be powerful tools for system change, as they provide patient-specific information to health professionals and organizations. Incorporating PROMs and PREMs data into patient care has been directly associated with improved care and improved effectiveness of treatments [1]. PROMs and PREMs enable patients to report on their quality of life, daily functioning, symptoms and other aspects of their health and well-being. Health services can also use this type of data to assist in measuring organizational performance and to determine the services and care that patients require and request [1].

Patient-reported experience measures (PREMs) are collected directly from patients, pertaining to their individual and unique experiences within the health service or their condition [2], while PROM data can relate to the patient’s perception of the impact of their condition (e.g., symptom severity, pain scales, function and/or health-related quality of life) or their interaction and satisfaction with the health service. Patients can input information on both disease-specific and general measures of function and health, which can be tracked over time [2]. Both PROMs and PREMs data can be obtained through questionnaires and/or surveys, with the aim of the collected data being used to initiate or support quality improvement activities and changes in health care [3]. Incorporating PROMs and PREMs data into clinical practice has the potential to narrow the gap between the clinician’s and patient’s views of the care provided and help tailor individualised care plans to meet the patient’s preferences and needs [4]. Evidence also exists identifying positive associations between patient safety and patient experience [1].

Despite the growing interest in and greater use of PREMs and PROMs, evidence suggests that barriers and challenges exist when collecting data, as well as in the implementation of changes based on the data collected. A systematic review by Gleeson et al. [3] examined 11 studies reporting on the use of patient experience data to inform improvement activities, suggesting that further research is required to identify aspects of the data that could enhance clinician engagement with using such data for future improvement activities [3]. Another systematic review of 22 randomized controlled trials by Ishaque et al. [5] identified 25 outcomes, of which only 18 were reported back to clinicians and used within patient care. The authors reported a focus on reporting statistical significance, with little attention paid to clinical significance or meaning for the patient or the service [5]. Boyce et al. [6] suggested that including clinicians when planning interventions may improve engagement in the use of PROM data. These reviews all highlight some of the discrepancies and challenges around using PROM data, which will be examined further in this review.

Valderas et al. [7] reported an earlier systematic review of the impact of PROMs data in clinical practice, highlighting a predominance of studies being conducted in the USA at the time. The results of the review also noted the methodological challenges of such studies and recommended that more rigorous work be conducted to assist clinicians to determine the feasibility and applicability of incorporating PROM data in practice. Sutherland et al. [8] also highlighted challenges when comparing PROM data across countries. Variation in data collection methods, measurement tools, data coding and follow-up were some of the limitations and issues identified in their cross-country comparative study. Collectively, these challenges are proposed to be some of the factors that impact the use of these types of data and may contribute to the gap in the uptake of data by clinicians and health services [2,4,9,10].

The objective of this scoping review was to review the international literature to collect and map the use of collected PREMs and PROMs data within acute healthcare contexts. More specifically, the concepts that this review aimed to explore included:What are the geographical contexts of PREMs and PROMs studies?What are the characteristics of this type of data?In what ways have collected patient-reported data been applied to practice in acute care settings?What barriers and facilitators have been identified in studies that used patient-reported data?

## 2. Methods

The scoping review was conducted in accordance with the JBI methodology for scoping reviews. A protocol for the review has been previously reported with detailed inclusion criteria [11]. Selected studies focused on patient-reported data that were proposed to facilitate and/or lead to organizational changes in service or care. We do not report on PROM data used for individual symptom or disease management.

### 2.1. Search Strategy and Study Selection

The search strategy aimed to find relevant literature published in the English language within the last 13 years from 2009 to 2021. As per the protocol [11], the types of studies included qualitative or quantitative primary studies, as well as systematic reviews and grey literature sources.

A three-step search strategy was used, with an initial limited search of Medline (via Ovid) and CINAHL (via EBSCO host) to identify relevant articles on the topic. This was followed by an analysis of the text words contained in the titles and abstracts, as well as the index terms used to describe relevant articles. A second search was then undertaken using all identified keywords and index terms across all included databases and registries. The following databases and registries were searched initially in August 2020 and updated in April 2021: PubMed (via MEDLINE), Embase (via EBSCO host) and CINAHL (via EBSCO Host). The trial registries searched included the Cochrane Central Register of Controlled Trials, Australia and New Zealand Clinical Trials Registry and the ISRCTN registry. The ProQuest Dissertations and Theses Global database was also searched for unpublished studies. The reference lists of all identified articles were searched for additional studies. The following keywords were used: PREMs, PROMs, patient-reported experience, patient outcome measures, patient satisfaction, acute care, tertiary care and hospital. An example of the Ovid MEDLINE search can be found in Appendix A.

Following the search process, studies were selected according to whether they met the review questions and concepts of interest for the scoping review. More specifically, these were:To identify the types of research studies that have been reported on the use of patient-reported data in acute hospital contexts;To map the characteristics of patient-reported data collected (related to service improvement), including geographical contexts;To identify the ways in which these types of data have been applied to clinical practice in acute hospital settings;To identify the barriers and facilitators that have been identified in studies reporting on the use of patient-reported data [11].

### 2.2. Modifications from Original Protocol

The search was completed according to the original protocol [11]; however, we specify here that we included papers using patient satisfaction surveys as patient experience measures, which provide data that drive change within hospital settings. Our search strategy was broad to accommodate this. We excluded searching the Australian Institute of Health and Welfare (AIHW), as this site reports on Australian health and welfare data, rather than international research studies on acute care health services. Due to the numbers of studies and different considerations for collecting PREMs/PROMs data in children [12], we decided to exclude studies pertaining to paediatric populations.

### 2.3. Data Extraction

Relevant data were extracted from the included studies to address the scoping review question using the methodology detailed by Peters et al. [13] and Khalil et al. [14]. A data collection tool was developed based on the concepts of interest and the research questions to enable the characteristics of the studies found to be charted [13]. More specifically, data were extracted into a template with the following headings: citation details, country of origin, methodology, data collection methods, measurement tools (where applicable), participants/sample details, sample size, setting, phenomenon of interest, results, conclusion, application to practice, barriers and facilitators.

### 2.4. Results

The electronic search of databases and registries identified 4057 records. Seven studies were added from hand searching. All records were exported from the databases into Endnote X9.3. A total of 77 records were deleted due to them being duplicates or not in the English language, with the remainder screened for eligibility. Two reviewers independently screened the records by title and abstract initially, with a third reviewer reviewing the updated search results. After screening, a total of 105 full articles were deemed to be eligible for full-text assessment. After full-text screening, 19 articles were excluded due to ineligible settings (*n* = 7), not being a research study (*n* = 3), ineligible population (*n* = 2) and being off-topic (*n* = 7). The search decision process is summarized in Figure 1 [15]. Extraction of the data was conducted by two reviewers as per the categories outlined above, with consideration of the main parameters needed to address the questions of the review.

## 3. Mapping the Results

The extracted data from single studies are presented in tables and figures to reflect the descriptive summary of the results aligning with the scoping review questions.

### 3.1. Concept 1: Geographical Contexts of Included Studies 

A total of 24 (27.9%) studies were from the United States of America [16,17,18,19,20,21,22,23,24,25,26,27,28,29,30,31,32,33,34,35,36,37,38,39], 15 (17.4%) from the United Kingdom [3,6,40,41,42,43,44,45,46,47,48,49,50,51,52], 8 (9.3%) from Canada [53,54,55,56,57,58,59,60], 6 (6.9%) from the Netherlands [61,62,63,64,65,66], 5 (5.8%) from Australia [5,67,68,69,70], 4 (4.6%) from Nigeria [71,72,73,74], 2 (2.3%) from Hong Kong [75,76], 3 (3.5%) from Ireland [77,78,79], 3 (3.4%) from India [80,81,82], 2 (2.3%) from Italy [83,84] and 2 (2.3%) from Greece [85,86]. Single studies (1.2%) were reported from Sweden [87], Germany [88], China [89], Denmark [90], Ethiopia [91], Pakistan [92], Uganda [93], Finland [94], Ghana [95], Tanzania [96], Saudi Arabia [97] and New Zealand [98]. One study was reported from across both Australian and New Zealand hospitals [99]. Distribution of publications is seen in Figure 2.

### 3.2. Concept 2: Characteristics of Included Studies

#### 3.2.1. Study Populations and Settings

The population sizes for the single studies included in this review ranged from 7 participants [40] to 2,648,275 [38]. The majority of studies (*n* = 28; 32.5%) were undertaken in general acute hospital inpatient units [16,17,19,21,22,27,30,31,32,35,36,39,42,44,45,57,64,68,75,76,80,83,85,86,87,89,92,95], while 12 studies (13.9%) were reported more specifically as being set within the Emergency Department [23,24,25,29,52,53,59,66,77,79,82,84] and 8 studies (9.3%) were relevant to surgical settings [20,26,28,33,47,58,78,98]. Thirteen studies (15.1%) were undertaken in outpatient clinics [34,38,43,55,60,71,73,74,81,91,94,96,97]. Three studies (3.4%) reported on inpatient and outpatients as participants [65,72,93] and three studies (3.4%) included data collected from health professionals [40,41,48]. Nine studies (10.4%) took place within oncology, haematology or palliative care inpatient or outpatient units [18,37,40,41,46,54,56,88,90]. Single studies reported on patients undergoing colonoscopy [67] and haemodialysis [49], and one paper reported on two patient groups with rare diseases of sclerosing cholangitis and kidney disease requiring transplant [48]. Some studies reported mixed settings/patient groups. Further details are provided in Figure 3.

#### 3.2.2. Study Design, Methodology and Data Collection Methods

Seven systematic reviews [3,5,6,50,51,61,62] were included as they contained aspects that pertained to the use of PREMs or PROMs data in practice and/or barriers and facilitators to the implementation of the collected data. Table 1 provides a summary of these papers. It is important to note that some of the included systematic reviews included studies of paediatric populations and/or settings other than hospitals. One systematic review protocol was also included in our selection [69], and the authors were contacted to see if the review was completed, but no data were available at this time. The review aims to examine factors influencing the fidelity of implementation of PROMs data and is ongoing.

Of the primary studies, eleven (12.7%) used qualitative methods [42,43,48,49,53,54,57,68,77,90,98], with one of these studies reporting an ethnographic approach [90]. Fifty-six studies (65.1%) reported using quantitative methods [16,17,18,19,20,21,22,23,24,25,26,27,28,29,30,31,32,34,35,36,37,38,39,44,45,47,52,55,56,58,59,60,64,65,66,67,70,71,72,73,74,75,78,79,80,81,82,83,86,87,88,89,93,96,97,99].

Eleven of the included papers (12.7%) used mixed methods for their studies [33,40,41,46,76,84,85,91,92,94,95]. Two of the quantitative studies (2.3%) reported specifically on using the Kaizen Lean methodology [59,78], which is based on the principles of process standardisation for quality improvement. Six studies (6.9%) examined various factors influencing patient satisfaction [35,38,79,82,89,93] and five other studies (5.8%) explored predictors of patient satisfaction [20,91] and/or correlations with other outcomes [26,44,58] (e.g., adverse events). We also included one clinical trial protocol in this review [49], as it outlined a prospective study that specifically examined the use of collected PROM data, as well as barriers and facilitators related to the data and preferred methods of feedback from both patients and physicians.

Regarding the data collection methods, surveys or questionnaires were the most popular method, as reported in 65 (75.5%) studies [16,17,18,19,20,21,22,23,24,25,26,27,28,29,30,31,32,33,34,35,36,37,38,39,40,41,44,45,46,47,52,54,55,56,58,59,60,64,65,66,67,71,72,73,74,75,76,78,79,80,82,83,84,85,86,87,88,89,91,92,93,94,95,96,97,99]. Focus groups were utilized in 6 (6.9%) studies [41,49,68,77,90,95] and interviews (including semi-structured interviews) were reported in 15 (17.4%) studies as data collection methods [33,40,42,46,48,49,53,57,77,81,84,85,91,92,98].

#### 3.2.3. Measurement Tools Used in Included Studies

The Hospital Consumer Assessment of Healthcare Providers and Systems tool (HCAHPS) was the most frequently used tool for measuring patient satisfaction, used either in its entirety or parts thereof [17,19,20,21,24,25,26,27,28,30,32,33,35,39,58]. Other frequently used tools included Press Ganey surveys for measuring patient satisfaction [33,38,39,66,68] and versions or variations of the 15-item Picker survey [45,46,55,75,84]. Table 2 presents the measurement scales used to collect patient experience and/or patient satisfaction data.

Validated scales were used to collect PROM data, particularly within oncology studies [37,40,41,56] and studies of renal patients [49,99] or those with rare diseases [48]. Most studies used non-validated tools that were developed by the authors based on the literature and/or previous research. Three studies reported on the development and psychometric properties of a new patient-report measurement tool [34,52,76]. Table 3 presents the measurement scales used for individual patient-reported data.

### 3.3. Concept 3: Application of PREMs/PROMs Data to Clinical Practice

Of the 86 included papers, only 22 primary studies reported on ways in which the collected data were used in practice, which are discussed in more detail below. Each of the six included systematic reviews included some detail pertaining to the application of data to clinical practice (refer Table 1). Two systematic reviews (total of 33 included studies) [3,61] addressed quality improvement programs arising from collected PROMs/PREMs data. One additional study [47] was reporting on an aspect of a larger PhD thesis, which examined a new model of care for supplying patient medication at discharge. Collected PROM data were used to support the new model. Roberts et al. [69] reported ongoing work to examine factors that influence the fidelity of the implementation of PROMs in routine patient care. 

The most frequently reported utilisation of the collected data was to improve communication between patients and health care staff. Two studies [24,29] reported on the use of the AIDET (Acknowledge, Introduce, Duration, Explanation, Thank you) communication framework, with results from their studies highlighting improvement in patient satisfaction scores after implementing the framework. Real-time feedback was reported on as a way to also improve patient–physician communication [21].

Four qualitative studies reported perspectives on the use of collected data [48,49,53,90]. Thestrup Hansen et al. [90] conducted an ethnographic study which explored patients’ perspectives on the use of PROMs data in haematology clinics. Patients reported that undertaking a PROMs survey provided them with topics to discuss with a doctor; however, some reported being confused by the purpose and utility of the data. Aiyegbusi et al. [48] reported patient and physician perspectives on the use of PROM data, with both groups reporting communication benefits through using PROM data to instigate discussions on quality of life or symptom management. It was also reported that the use of PROMs data facilitated patient-centred care and patient involvement, although it was suggested by physicians that specific PROM tools were more effective than generic tools for their patient cohort who had rare diseases [48]. Dainty et al. [53] reported physicians’ perspectives on the use of PROM data in the Emergency Department (ED) and identified tension between the dynamic and complex nature of the ED and the application of PROM data. More specifically, participants suggested that, as they often saw patients for short periods and frequently at times of crisis, there was preference for the use of objective data (e.g., detail on adverse events and readmission data). Concerns were voiced regarding the legal and ethical implications of using patient-reported data due to limited follow-up with patients. Other issues of the subjectivity and timing of collecting data were also seen as limitations to the application of PROMs in the ED; however, some participants suggested a benefit in using PROMs to facilitate communication between patients and physicians. The authors highlighted the importance of physician involvement in planning and implementing PROMs in the ED [53]. Anderson et al. [49] reported a clinical trial protocol for a qualitative study on the use of PROMs in patients requiring haemodialysis. This work is ongoing and will capture data on the use of electronic PROMs for this context.

Other applications of collected data included the establishment of a patient liaison program to monitor patients’ satisfaction and improve care and patient flow within the emergency department [23,59], an increase in screening rates for patient distress within the oncology setting [37], the implementation of nursing care bundles [32], strategies for reducing noise levels within hospital wards [33] or improving cleanliness and waiting times in the outpatient department [96]. The implementation of specialist oncology nurses [40,41] or a nurse–midwife [36] to coordinate patient care and flow through different care settings was also reported on and evaluated using patient-specific or general satisfaction surveys. Additionally, data were used to reorientate care towards patient’s centeredness to ensure that patients are the focus of their care and participate in their own goal setting [22].

Multifaceted interventions were reported on in two additional quality-improvement projects [16,30], whereby audit data were collected pre- and post-interventions, with improvement noted in both patient satisfaction scores and staff responses. Improvements addressed (but were not limited to) patient–staff communication, patient discharge instructions, patient nutrition, staff availability and maintenance of patient dignity.

### 3.4. Concept 4: Facilitators and Barriers to Using Collected PREMs and PROMs Data

Several studies reported on barriers to and facilitators of collecting PROMs data; however, for this concept, we report on barriers to and facilitators of using or implementing collected PROMs/PREMs data in hospitals for changes in services or practice. Of the systematic review papers, Greenhalgh et al. [51] highlighted the distinction between individual PROMs measures (such as those for symptom reporting or pain scales) and measures that are used for service improvement or service quality measures (e.g., satisfaction scores). Boyce et al. [6] reported on sixteen studies in their review, exploring health professionals’ experiences of using collected data, with the barriers reported on including inadequate resources and poor attitudes to PROM use. Although participants inferred benefit to the use of collected PROMs data for improving patient care, concerns were raised around patient privacy, confusion regarding the goals or aims of implementing PROMs data and limited managerial support to make changes. Incorporating clinicians at the planning stage of any PROM-based intervention was recommended to facilitate change [6].

Forster et al. [50] reported on six systematic reviews (118 total studies), with a focus on facilitators of and barriers to implementing patient-reported outcome measures for health services. Most of the included studies reported on challenges associated with collecting data, rather than the use of collected data, and the authors recommended further research on how organisational culture impacts and supports changes in practice following the collection of PROMs data. Across the relevant studies, barriers and facilitators were reported under patient, staff organisational or data and intervention-related categories. A summary of the barriers to and facilitators of implementing data to effect change in service or patient outcomes, according to these categories, is seen in Table 4.

## 4. Discussion

This scoping review mapped several characteristics of studies that report on patient-collected data and identified how the data are being used specifically within hospitals. We found that most studies were undertaken in developed countries, and a majority were focused on general satisfaction with hospital care or processes. A diverse range of measurement tools were identified, with less than half of the included studies measuring the implementation of a specific intervention or a quality improvement program following from data collection.

PROMs and PREMs data were developed originally to measure disease burden and provide a mechanism for healthcare providers to prioritise areas of need as reported by their patients [100]. A move towards using patient-reported data to address organisational needs has increased the range and scope of patient-collected data and has led to questions regarding the use of collected data. The findings of this review have identified few studies reporting on the use of collected data in hospital settings to effect change in care or service delivery. This supports other commentary on the gap between collecting and using such data in practice [101,102]. Global variation in health care systems is a consideration when examining the results of this review. The literature highlights that most studies were reported from developed countries, which may reflect variations in health priorities, resource availability, or other factors that have not been captured in this review. Further exploration of the variation in the use of patient-collected data across different types of health services would be beneficial.

Improving patient experience has been associated with decreased cost of care and reduction of health services utilisation [1]. Quantitative and qualitative patient-reported surveys are commonly used within hospitals to reflect an individual’s experience with the engagement of the service; however, challenges have been noted in analysing large amounts of qualitative data, or, conversely, placing appropriate value on quantitative results to accurately capture the patient perspective [3]. The adoption of external agencies to collect this type of data is common, but it is suggested that this can affect the utility of findings for the service due to differing priorities. In their review on the use of patient-reported experience data, Gleeson and colleagues identified that capturing patient experience data to facilitate improvement in care is challenging. Recommendations from their systematic review highlighted the need for appropriate resources, expertise and leadership support if patient-collected data are to be analysed and implemented appropriately within hospital settings [3].

Of the few studies that reported on using PREMs and PROMs data to improve patient care, successful changes were reported when leadership and organizational support were strong. Building strong positive relationships between staff and managers has also been reported to improve workplace culture and subsequent patient experience and well-being [87,103,104]. Our review has further highlighted barriers and facilitators that can be employed when developing and implementing strategies that arise following the collection of patient-reported data.

PROMs and PREMs tools can be used to gather data at individual, group and organisational levels. The use of data for subsequent quality improvement strategies has been reported as complex and challenging [100,105], and is often attributed to methodological issues with the data collection process. The subjectivity of participant responses, use of unvalidated tools, the variety of tools being used and limitations of patients’ understanding of the tools are some issues reported to influence the collection and subsequent use of patient-reported data. [106] Nevertheless, these types of measures are integral to obtaining patient perspectives and improving care delivery through providing person-centred care. The use of standardized tools, ensuring that the correct measurement tool is being used to address a clear aim and selecting the appropriate tool for the context in which it is being used [50] are essential considerations to support considerate and relevant use of patient-reported data.

## 5. Strengths and Limitations

This review was conducted using a rigorous framework and following the publication of a developed protocol [11]. A broad search was conducted of the international literature; however, due to the inconsistency of the nomenclature in the PREMs/PROMs literature, it is feasible that studies were overlooked. The use of various terminologies for PREMs and PROMs data has been reported as a major barrier to the consistent application of findings. To overcome this limitation, we conducted a broad search and included studies that also reported on the use of patient satisfaction surveys. We also asked a health librarian to assist with developing the search and searched multiple databases and repositories for published and unpublished studies. In the future, the use of standardised terminology would support the inclusion of a wider range of studies.

Inconsistent terminology has also impacted categorising the data and, as such, some studies that were more general in focus may have not been reported on in as much detail. Other challenges were noted in how study designs were reported including limited demographic detail. Several included reports were in abstract form and were, therefore, limited in the data available; however, it was important to include these to represent the current research being undertaken. Adherence to reporting guidelines (e.g., COSMIN) for patient satisfaction and PREMs/PROMs studies would assist with reporting consistency.

Finally, our review did not include paediatric studies and was only focused on hospital settings. Other health care settings and populations could be a focus for future scoping reviews on this topic.

## 6. Conclusions

This scoping review reported the characteristics of the current research literature on the use of collected PREMs, PROMs and patient satisfaction data in acute care contexts. The results indicate that these types of studies are more commonly found in developed countries and within oncology settings. This review also highlights the widespread nature of collecting patient-reported data and raises further questions on how patient-reported data are best used within this setting. Consistency of terminology, use of standardized measures and differentiating between individual or organisational patient-reported data are important considerations for future research in this field.

## Figures and Tables

**Figure 1 ijerph-19-11160-f001:**
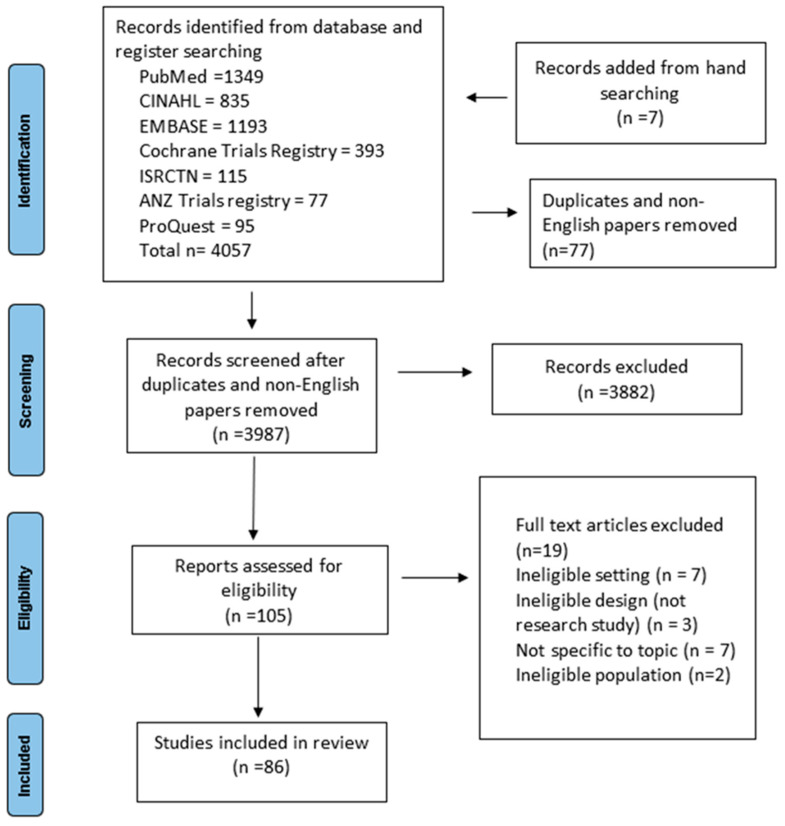
PRISMA flowchart of the study selection and inclusion process [15].

**Figure 2 ijerph-19-11160-f002:**
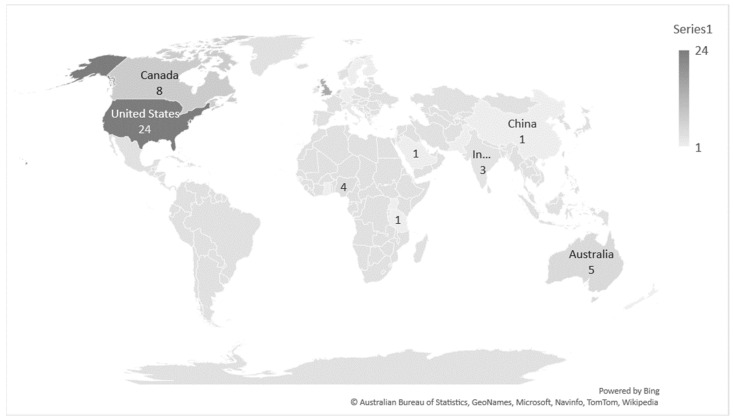
Geographical distribution of the included papers.

**Figure 3 ijerph-19-11160-f003:**
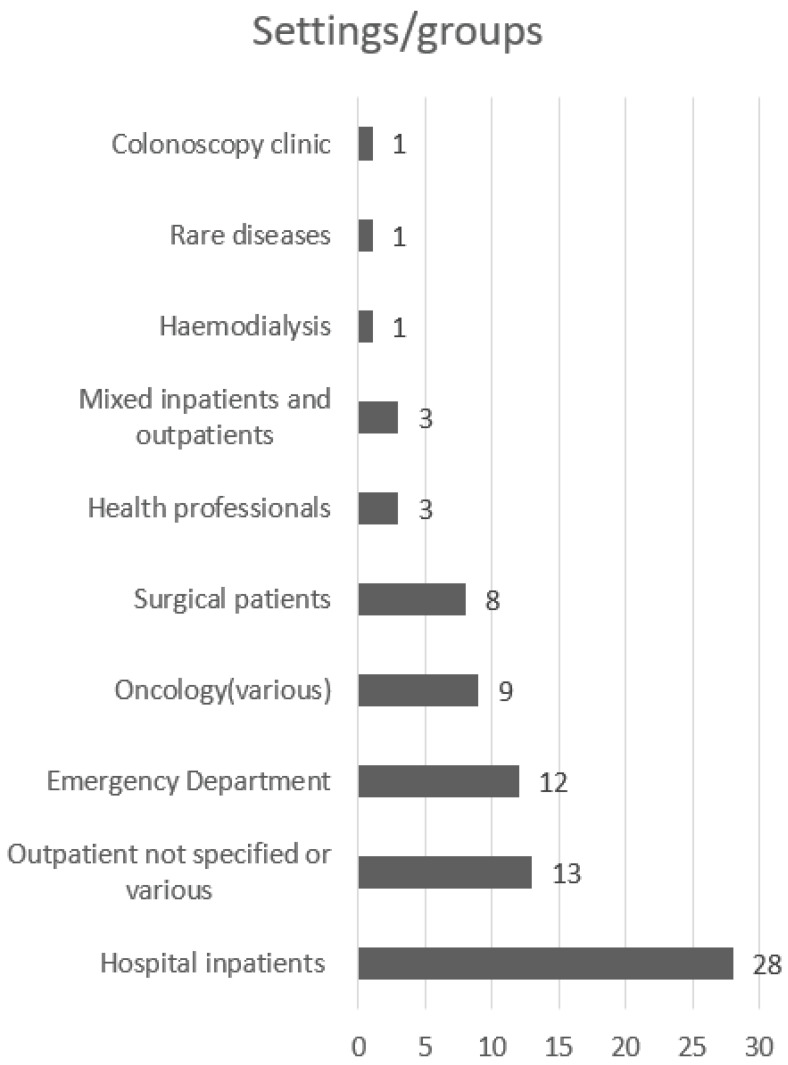
Study settings/contexts (*n* = 79).

**Table 1 ijerph-19-11160-t001:** Characteristics of systematic reviews relevant to the topic.

Author	Country of Origin	Number of Included Studies	Type of Review and Primary Focus	Summary of Results (Pertaining to Implementation)
Bastemeijer, et al., 2019 [61]	Netherlands	21	Systematic review (SR) of studies that reported on quality improvement activities in hospital settings based on patient experience data; barriers and promoters also reported on	Quality improvement (QI) strategies included staff and patient education, audit and feedback processes, clinician reminders, organisational and policy change. Barriers pertain to data collection, lack of time and scepticism regarding the benefits of change. Organisational support staff and patient involvement were reported as facilitators
Boyce et al., 2014 [6]	UK	16	Qualitative synthesis examining the experience of health professionals using PROM data to improve care quality	Barriers to and facilitators of the use of data were reported under four themes, summarised as practical, attitudinal, methodological and impact categories. Infrastructure, timing and workload must be considered prior to collecting PROM data to ensure the use of findings. Including staff in the planning stage may improve engagement, attitudes and subsequent use of data. Interpretation of PROMs data varies, which requires further consideration
Foster et al., 2018 [50]	UK	6 systematic reviews	Systematic review of systematic reviews	Time and resources needed for preparing and designing processes for implementing PROMs and changes relating to PROMs data. Recommendation for ‘leader’ to facilitate the implementation of strategies based on feedback. Contextual considerations and staff training are needed. Gaps identified in factors that influence the implementation of PROMs
Gleeson et al., 2016 [3]	UK	11	SR of how PREMs are collected and used to inform QI projects in hospitals; barriers to and facilitators of using patient experience data also reported	Patient experience data mostly collected via surveys. Difficulties noted in evaluating any changes from implementing the results of experience data into practice. Formal staff training suggested for the analysis of data and implementation of subsequent QI projects, as a lack of confidence in interpreting data was seen as a major barrier
Graupner et al., 2021 [62]	Netherlands	22	SR of the effectiveness of PROMs on patient outcomes, patient experiences and process indicators in cancer care. Fifteen studies compared PROMs with no PROM	Feedback to health professionals and patients from collected PROM data led to improvements in symptom management, communication between patients and healthcare providers, as well as HRQoL and patient satisfaction. Results were not statistically significant due to small samples
Greenhalgh et al., 2017 [51]	UK	36	Two realist syntheses; one to develop a classification and taxonomy of programme theories with the development of a logic model on the collation, interpretations and use of PROMs data. The second synthesis explored how PROMs data work in practice in detail, including (but not limited to) an analysis of barriers and supporters of the implementation process and unintended consequences	PROMs data that were deemed to be clear and credible, focused on patient care improvement and that were timely, were more likely to be used to develop improvement strategies. System-wide approaches were then needed for implementing improvement strategies. PROMs were a beneficial method for patients to raise concerns, but improvements in communication with health care providers were less overt. The reviews highlighted challenges with moving beyond collecting PROM data to effectively using results for any change in practice
Ishaque et al., 2019 [5]	Australia	22 studies included with 25 comparisons	SR of RCTs comparing the effectiveness of PROM with no PROM, with outcomes including health care processes, health outcomes and satisfaction with care	Improvements noted in clinician/patient communication and decision making; however, many studies focused on statistical significance, rather than highlighting clinically meaningful changes in outcomes or care processes. Some, but not all, studies implemented strategies based on PROMs use. Methodological limitations noted within studies

**Table 2 ijerph-19-11160-t002:** Tools cited within the included primary studies measuring patient satisfaction with care or quality of service within acute settings.

Measurement Tool	Cited Study
MedRisk Instrument for Measuring Patient Satisfaction with Physical Therapy Care (MRPS)	Algudairi et al. [97]
Picker Patient Experience Questionnaire-15 (PPE-15) (or adaptation of)	Andres et al. [75]
	Forman et al. [55]
	Robinson et al. [25]
	Seghieri et al. [84]
	Tsianakas et al. [46]
A&E department questionnaire	Bos et al. [52]
HCAHPS (Hospital Consumer Assessment of Healthcare Providers Survey)	Figueroa et al. [17]
	Gupta et al. [19]
	Iannuzzi et al. [20]
	Indovina et al. [21]
	Otani et al. [35]
	Prabhu et al. [58]
	Sacks et al. [26]
	Seiler et al. [27]
	Shirk et al. [28]
	Siddiqui et al. [39]
	Smith et al. [30]
	Stanowski et al. [31]
	Trail-Mahan et al. [32]
	Wilson et al. [33]
Ambulatory Oncology Patient Satisfaction Survey (AOPSS)	Fitch et al. [54]
Quality from the Patient’s Perspective (QPP) questionnaire	FrÖjd et al. [87]
Press Ganey Survey	Fulton et al. [38]
	Louis et al. [66]
	Rapport et al. [68]
	Siddiqui et al. [39]
Healthcare Climate Questionnaire (heard and understood root question)	Gramling et al. [18]
Patient Perceptions of Patient-Centeredness Questionnaire	Gramling et al. [18]
FAMCARE (caregivers) and FAMCARE-Patient (patients) scales	Hannon et al. [56]
Intensive Care Experience Questionnaire (ICEQ)	Kelepouri et al. [85]
Core Questionnaire for the Assessment of Patient satisfaction (COPS)	Kellezi et al. [42]
Client Satisfaction Questionnaire (CSQ)	O’Regan and Ryan [79]
Quality for the Patient’s Perspective of ED	Preyde et al. [59]
Leeds Satisfaction Questionnaire (LSQ)	Shu et al. [60]
Inpatient Assessment of Health Care (I-PAHC) modified from CAHPS instrument	Sipsma et al. [89]
SERVQUAL Questionnaire	Umoke et al. [74]
Hong Kong Inpatient Experience Questionnaire (HKIEQ)	Wong et al. [76]
Modified SERVAL form	Ogunnowo et al. [73]
Clinically Useful Patient Satisfaction Scale (CUPSS)	Zimmerman et al. [34]
Validated but unknown patient satisfaction survey	Lim et al. [23]
	Matis et al. [86]
	Rajendiran et al. [82]
Tool developed by researchers (based on prior research or the literature +/− pilot testing or validation)	Bhaskar et al. [80]
	Geberemichael et al. [91]
	Iloh et al. [71]
	Jiang et al. [22]
	Kamiya et al. [96]
	Kanwal et al. [92]
	Morton et al. [99]
	Nabbuye-Sekandi et al. [93]
	Obi et al. [72]
	Paul et al. [36]
	Puri et al. [81]
	Ullah et al. [78]
	Wright et al. [47]
Hospital-developed survey	Lin et al. [67]
	Raleigh et al. [44]
	Werkkala et al. [94]
	Yawson et al. [95]
Unspecified/other surveys	Robertson et al. [24]
	Ruggieri et al. [83]
	Skaggs et al. [29]

**Table 3 ijerph-19-11160-t003:** Tools cited within the included primary studies for measuring individual patient-reported outcomes within acute settings.

Measurement Tool	Cited Study
Chronic Liver Disease Questionnaire (CLDQ) Medical Outcomes Study—Short Form 12 (SF-12) Paediatric Quality of Life Inventory—Transplant Module version 3.0 (PedsQL-TM 3.0) * EuroQoL-5 dimensions (EQ. 5D)	Aiyegbusi et al. [48]
European Organisation for Research and Treatment of Cancer (EORTC) QLQ-C30	Lamprecht et al. [88]
Kidney Disease Quality of Life—36 (KDQOL-36) Kidney Disease Quality of Life—SF (KDQOL-SF) Integrated Patient Outcome Scale—Renal (IPOS-Renal)	Anderson et al. [49]
National Comprehensive Cancer Network Emotional Distress Thermometer (EDT) paper tool	Chiang et al. [37]
Caregiver QOL Index—Cancer Functional Assessment of Chronic Illness Therapy—Spiritual Wellbeing Scale (FACIT-Sp)—inclusive of functional assessment of cancer Therapy–General (FACT-G) and 12-item Functional Assessment of Chronic Illness Therapy—Spiritual Well-Being (FACIT-Sp-12)	Hannon et al. [56]
Supportive Care Needs Survey—Short Form 34 (SCNS-SF34) Problems Checklist Cancer Needs Questionnaire—Short Form Psychosocial Needs Inventory Comprehensive Needs Assessment Tool in Cancer (CNAT) Cervical Cancer Concerns Questionnaire (CCCQ) Cancer Rehabilitation Evaluation System—Short Form Cancer Needs Distress Inventory	Kotronoulas et al. [40]
Supportive Care Needs Survey—Short Form 34 (SCNS-SF34) Problems Checklist Cancer Needs Questionnaire—Short Form Psychosocial Needs Inventory Cancer Survivors Unmet Needs Functional Assessment of Cancer Therapy—Colorectal concerns subscale (FACT-C)	Kotronoulas et al. [41]
European Organisation for Research and Treatment of Cancer (EORTC QLQ-C30)—Danish version Outcomes and Experiences Questionnaire (OEQ)	Thestrup-Hansen et al. [90]
Domains of the PROMIS profile 29.0 (Dutch Version)	van Galen et al. [64]

* This tool was used for patients aged 16–25.

**Table 4 ijerph-19-11160-t004:** Barriers to and facilitators of the use of collected PROM/PREM data.

	Barriers	Facilitators
Patient [3,47,59]	Patient understanding of measures impacting confidence in data Language barriers Unrealistic or unclear expectations	Incorporate patient representatives Incorporation of patient views Patient involvement
Staff [3,16,29,32,33,40,47,53,59]	Ineffective communication Change fatigue Limited staff time Staff attitudes Unclear expectations Staff availability Staff perceptions of burden Lack of understanding Lack of expertise Staff’s limited understanding of data analysis to be able to use results Staff resistance to change	Purposeful leadership rounding Staff education Leadership support Staff education Educating new staff Including all health professionals in Intervention to ensure shared and effective communication Collaboration Dedicated time for managerial staff to support and supervise information sharing Clear aims and purpose Redistribution of roles Training Dedicated time for staff encouragement Regular meetings
Organisational [3,16,29,32,33,47,53,59,61]	Overlapping initiatives impacted the certainty of improvement results Unclear expectations Cost Lack of time and limited resources Competing priorities Financial goals Organisational culture	Reminders of hospital goals Committee establishment Leadership support Supportive culture Culture change Celebration events Organisational support Culture of improvement Management support
Data/Intervention related [3,7,29,32,33,41,47,53,59]	Confounding factors Inconsistency in data collection Time-intensive intervention with limited resources Unclear if actual intervention improved post-implementation data or other effects due to time taken for implementation Timing of feedback Lack of specificity Relevance of data-collection tool Number of patients needed for sample	Random audits Availability of real-time data Incorporate clinicians at planning stage Understanding of context Use of technology Use of validated tools Context-specific surveys Public reporting of results

## Data Availability

Not applicable.

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
