# Peer review of "Use of Patient-Reported Data within the Acute Healthcare Context: A Scoping Review"

_ijerph, 2022, doi:10.3390/ijerph191811160_

Round 1
Reviewer 1 Report
Thanks to the authors for the opportunity to review this work. I found it interesting, but there are some aspects I would like to point out:
1. Introduction seems to be a succession of authors' statements, more appropriate to the discussion. The authors should focus more on the importance of PREMs and PROMs, current use, perspectives...
2. Methodology is adequate, but 7 systematic reviews are mentioned that were judged to be relevant. Why? What criteria were used to qualify them as such?
3. There is a lack of integration of the results. It is not enough just to present the results, they must be integrated to give the reader a global perspective of the current situation of PREMs and PROMs.
4. Discussion is brief and does not put the research in context.
5. Conclusions do not follow from the objectives.
Author Response
Thank you to reviewer 1 for your comments on our manuscript and for the time spend reviewing the paper. We have endeavoured to address all feedback but would be happy to make further revisions as needed. Please see attached file for our responses.

Reviewer 2 Report
It's a well-written scoping literature review focusing on patient reported outcome measures and experience measures.
I would like the authors to describe thoroughly the criteria used in the screening process (include/exclude) there was a significant reduction in the sample (3987->105). It is the most crucial part of the study selection and could potentially change the outcomes of this literature review.
I don’t understand the reasoning of the sentence: “We excluded searching the Australian Institute of Health and Welfare (AIHW) due to the desire to include international research studies rather than country-specific health data.” In the article, there are many country-specific studies in the sample.
Why
Some minor remarks:
2009-2021 = 13 years
There is more like 2009-2020 ("updated in April 2021")
(lines 89-90) “The search strategy aimed to find studies published in English language, within the
last 11 years from 2009 to 2021.
No value (“15”) for the United Kingdom in Figure 2
Author Response
Thank you to reviewer 2 for your time reviewing our paper and your feedback. Please see attached file for our responses. We are happy to make any further changes as needed.

Reviewer 3 Report
Introduction: Ok, recent bibliography
2. Methods: The concept of acute medicine implies a very broad field of analysis, which in turn implies more heterogeneity in your results. Have you found standardized tools for different acute medicine services?
Measurement Tools Used in Included Studies: (190-213)
Need to consider that the variability in the measurement tools used to collect PROM data may lead to greater heterogeneity in their results. could they be standardized?
Discussion:
They have to consider that in the developed countries where the studies have been done they have different health systems and this may condition the results.
Discussion:
"elucidated how the data is being used specifically within hospitals" (298) They should consider that such a broad study framework makes it difficult to analyze and obtain conclusions applicable to reality.
Author Response
Thank you to reviewer 3 for your time and feedback on our manuscript. We have endeavoured to respond to all comments but would be happy to make further revisions as necessary. Please see attached file.

Round 2
Reviewer 1 Report
Thank you for the opportunity to review your interesting manuscript
Author Response
Thank you to reviewer 1 for the time spent in reading through our manuscript and the positive comments